# Mitigating the impact of COVID-19 on primary healthcare interventions for the reduction of under-5 mortality in Bangladesh: Lessons learned through implementation research

**Alemayehu Amberbir**[1] *, **Fauzia A. Huda**[2], **Amelia VanderZanden**[1], **Kedest Mathewos**[1], **Jovial Thomas Ntawukuriryayo**[1], **Agnes Binagwaho**[1], **Lisa R. Hirschhorn**[1,3]

**1** University of Global Health Equity, Kigali, Rwanda, **2** International Centre for Diarrhoeal Disease Research, Bangladesh (icddr,b), Dhaka, Bangladesh, **3** Feinberg School of Medicine, Northwestern University, Chicago, Illinois, United States of America

* aleamberbir@gmail.com, aamberbir@ughe.org

**Data Availability Statement:** The data that support the findings of this study are available from the

## Abstract

The COVID-19 pandemic posed unprecedented challenges and threats to health systems, particularly affecting delivery of evidence-based interventions (EBIs) to reduce under-5 mortality (U5M) in resource-limited settings such as Bangladesh. We explored the level of disruption of these EBIs, strategies and contextual factors associated with preventing or mitigating service disruptions, and how previous efforts supported the work to maintain EBIs during the pandemic. We utilized a mixed methods implementation science approach, with data from: 1) desk review of available literature; 2) existing District Health Information System 2 (DHIS2) in Bangladesh; and 3) key informant interviews (KIIs), exploring evidence on changes in coverage, implementation strategies, and contextual factors influencing primary healthcare EBI coverage during March–December 2020. We used interrupted time series analysis (timeframe January 2019 to December 2020) using a Poisson regression model to estimate the impact of COVID-19 on DHIS2 indicators. We audio recorded, transcribed, and translated the qualitative data from KIIs. We used thematic analysis of coded interviews to identify emerging patterns and themes using the implementation research framework. Bangladesh had an initial drop in U5M-oriented EBIs during the early phase of the pandemic, which began recovering in June 2020. Barriers such as lockdown and movement restrictions, difficulties accessing medical care, and redirection of the health system's focus to the COVID-19 pandemic, resulted in reduced health-seeking behavior and service utilization. Strategies to prevent and respond to disruptions included data use for decision-making, use of digital platforms, and leveraging community-based healthcare delivery. Transferable lessons included collaboration and coordination of activities and community and civil society engagement, and investing in health system quality. Countries working to increase EBI implementation can learn from the barriers, strategies, and transferable lessons identified in this work in an effort to reduce and respond to health system disruptions in anticipation of future health system shocks.

Directorate General of Health Services under the Ministry of Health and Family Welfare in Bangladesh (requests may be sent to dg@ld.dghs. gov.bd and adgadmin@ld.dghs.gov.bd). However, restrictions apply to the availability of these data, which were used for the current study, and so are not publicly available. The datasets used and/or analyzed during the current study are available from the corresponding author on reasonable request and with permission of the Directorate General of Health Services under the Ministry of Health and Family Welfare in Bangladesh.

**Funding:** This work was supported by Gates Ventures (AA, AB, LRH). The funders had no role in study design, data collection and analysis, decision to publish, or preparation of the manuscript.

**Competing interests:** The authors have declared that no competing interests exist.

## Background

Since the beginning of the COVID-19 pandemic, there have been reports of disruptions in access to and availability of a number of the evidence-based interventions (EBIs) known to reduce under-5 mortality (U5M) across different countries [1–3]. Further, studies have shown indirect effects of COVID-19 on child mortality in low- and middle-income countries related to changes in the availability and affordability of health commodities [1, 2], as well as interruptions in healthcare and social protection services [3]. In the early days of the pandemic, it was predicted that COVID-19 could cause considerable disruptions to essential health services including maternal and child health services in Bangladesh [4]. Models estimated that the pandemic could deprive 1,654,500 children from receiving antibiotics for pneumonia treatment, 3,673,600 children from diphtheria-pertussis-tetanus immunization (DPT), and 386,500 pregnant women from accessing healthcare facilities for delivery, potentially resulting in an increase in child mortality by as much as 37% over the following year [4, 5]. Many feared that these health systems impacts would reverse the progress made towards the achievement of the Sustainable Development Goals in health. However, the drop in the U5M-focused EBIs has been variable, and there is a need to understand what strategies and contextual factors are associated with preventing or mitigating disruptions at the national and subnational levels in the delivery, quality, and equity of EBIs known to reduce U5M.

In Bangladesh, the first cases of COVID-19 were confirmed on March 8, 2020 and the first death on March 18, 2020 [6, 7]. On March 23, the government announced all government and private offices would close for 10 days from March 26 through April 4, 2020 [8]. The nationwide lockdown eventually extended through April 25, 2020 [9]. By December 27, 2020, there had been 509,148 confirmed COVID-19 cases and 7,452 deaths from COVID-19 in Bangladesh, with a case fatality ratio of 1.46% [10]. (See Fig 1 for a detailed timeline.) The Ministry of Health and Family Welfare leads Bangladesh's multifaceted health system and oversees two directorates: The Directorate General of Health Services (DGHS), primarily charged with

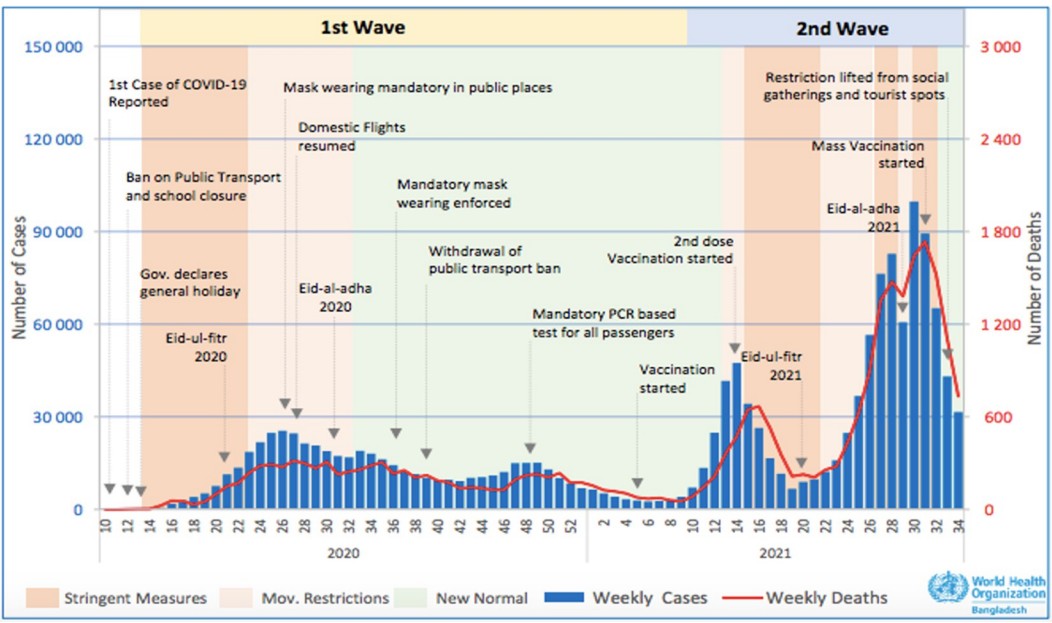

**Fig 1. Bangladesh COVID-19 timeline.** Source: WHO Bangladesh COVID-19 Morbidity and Mortality Weekly Update. 30 August 2021.

managing public health services and providing general healthcare; and the Directorate General of Family Planning (DGFP), initially responsible for managing and providing family planning and components of maternal healthcare. Primary healthcare interventions including those contributing to essential health services for children under 5 take place under both director-ates. Broad disruptions to access to essential health services in the first months after Bangla-desh's first cases of COVID-19 were confirmed; during the lockdown, many services were suspended [11]. There were substantial declines in access to medicines, treatments, and rou-tine medical care [12].

We previously conducted implementation research to understand how exemplar countries, like Bangladesh, outperformed their regional and economic peers in reducing U5M during the period 2000–2015, focusing on the implementation of EBIs known to reduce amenable U5M [13]. We examined the EBIs, implementation strategies, and contextual factors that either facil-itated or hindered this progress through an implementation research framework that we devel-oped and tested as part of this work [14]. Building on our previous work in Bangladesh [13], we designed a study to explore if and how health system-delivered EBIs targeting amenable U5M were maintained during the period of COVID-19 in Bangladesh. We explored what implementation strategies were used to prevent or respond to primary healthcare EBI drops, and the contextual factors that helped or hindered the response to COVID-19-related threats to EBI uptake and delivery. We considered how decisions were made to continue existing strategies used in earlier success in EBI implementation, what adaptations were made, and what new strategies were put into place in the country. We have put forward lessons for other countries still working to increase EBI implementation in future health system shocks.

## Methods

### Ethics statement

This study protocol was reviewed and approved by the Rwandan National Ethics Committee (Approval No: 1121/RNEC/2020), and the icddr,b Research Review Committee and (PR#21020, Date: 22 March 2021) Ethical Review Committee (PR#21020, Date: 14 April 2021). Written informed consent was obtained from individual key informants and anonymity and confidentiality were maintained throughout the project. For qualitative data, no quotes or spe-cific viewpoints which are identifiable to the individual source were included in the dissemina-tion products.

### Study design and approach

The study utilized a sequential explanatory mixed methods implementation science approach, using coverage data to inform qualitative questions [15]. The work was guided by the previ-ously used implementation research framework (see S1 Fig) [16]. This framework was designed during our previous work in Bangladesh and is a hybrid of existing implementation science frameworks including the Consolidated Framework for Implementation Research, Proctor's implementation outcomes, and modification from Aarons and colleagues' work [17–19]. The framework focused on specific stages of EBI implementation, adding in the critical, explicit step of Adaptation to the Exploration, Preparation, Implementation, and Sustainability framework (EPIAS) [16]. We utilized this framework to understand strategies and contextual factors at the different levels including global, national, subnational, facility, and community, that were implemented during the first phase of the COVID-19 pandemic period to prevent or respond to drops in primary healthcare EBI access, availability, or delivery. We explored the existing, adapted, or newly adopted strategies used, and we identified contextual barriers and facilitators responsible for either increased or decreased use and provision of EBIs known to

reduce amenable U5M. We used the Standards for Reporting Implementation Studies (StaRI) and COnsolidated criteria for REporting Qualitative research (COREQ) Checklists where relevant and they are included in the supplemental data (S1 and S2 Checklists).

## Data collection and measures

We collected data from three sources: desk review of available publications, reports, and policies; existing routine data from the District Health Information System 2 (DHIS2) of the DGHS under the Ministry of Health and Family Welfare (MoH&FW); and key informant interviews (KIIs), to focus on evidence on changes in coverage, implementation strategies, and contextual factors influencing primary healthcare EBI coverage. The study, which took place between February to October 2021, covered the period between January 2019 to December 2020.

**Desk review.** We conducted a desk review of available reports, policies, and peer reviewed publications, published between March 2020 and September 2021, accessed through PubMed and Medline, MOH policy documents and reports, and documents from WHO, Global Fund, Global Financing Facility, and other sources, to explore strategies and contextual factors as well as change in the delivery of specific EBIs used to reduce U5M, during the COVID-19 pandemic period March 2020 to December 2020. The desk review provided information on response and strategies associated with preventing, mitigating, or responding to disruptions in EBIs known to reduce amenable U5M in Bangladesh and other countries. We also reviewed gray literature which included MoH&FW reports for COVID-19 in Bangladesh, government response guidelines for COVID-19, as well as other sources including WHO guidelines, WHO Pulse Survey reports, and World Bank reports.

**DHIS2 data.** To understand whether Bangladesh maintained key primary healthcare EBIs for children under 5 during the COVID-19 pandemic, we analyzed existing DHIS2 data of the DGHS under the MoH&FW. The DHIS2 is Bangladesh's web based data collection system which routinely collects nationwide health data from public healthcare facilities. We explored this data disaggregated by time course over one year prior to COVID-19 (January 2019 to February 2020) and during the COVID-19 period (March 2020 to December 2020).

**Key informant interviews.** We adapted the codebook from the previous Bangladesh case study [11], based on emerging lessons from COVID-19 response and strategies which were associated with preventing, mitigating, or responding to disruptions in EBI access, uptake, and coverage, as well as the resiliency framework from Kruk and colleagues [20]. Between April to September 2021, we conducted 13 KIIs (led by FAH) via a videoconferencing platform with purposively sampled policymakers, donors, implementing partners, and health services providers in Bangladesh. The interviews, which typically lasted 40 minutes, were designed to identify implementation strategies and how they were selected, adapted, or newly adopted. The key informants helped to identify additional evidence of implementation strategy success or lack thereof, and explore why some strategies were found to be effective in other countries but not used in Bangladesh to prevent or respond to COVID-19-related drops in access and uptake of EBIs for the reduction of amenable U5M. Additionally, the KIIs explored the process by which decision-making was made and key contextual factors which were barriers, facilitators, or otherwise influenced decision-making.

## Data analysis

**Quantitative.** We analyzed quantitative data to describe changes in EBI uptake and delivery. We used scatter plots using interrupted time series analysis (timeframe from January 2019 to December 2020). We used interrupted time series segmented regression analyses by fitting a

Poisson regression model to estimate the impact of COVID-19 on childhood immunization, maternal health services (antenatal care visits and facility based delivery) and outpatient attendance for children. The model included a time variable, a dummy COVID-19 variable indicating pre-COVID-19 (January 2019 to February 2020) and the period of COVID-19 (March 2020 to December 2020), and a primary health service reporting calendar month. We adjusted the model for seasonality using two pairs of sine and cosine terms (Fourier terms) included in the model. These are pairs of sine and cosine functions of time with an underlying period reflecting the full seasonal cycle (i.e. calendar year). We analyzed the data using Stata 16.1 (Stata-corp, College Station Texas, USA).

**Qualitative.** We extracted desk review data from included studies and reports. As feasible, we explored contextual factors and new strategies used by Bangladesh and in the WHO region to maintain EBIs known to reduce U5M in the context of COVID-19. We audio-recorded the qualitative data from KIIs, transcribed it in the local language (Bangla) then translated where necessary, and entered it into Dedoose (Dedoose v-9.0.17). Two coders (AA and KM) reviewed memos through an iterative coding in Dedoose, and we used code weighting to describe relative frequency of categories. We used thematic analysis of coded interviews to identify emerging patterns and themes from the interviews [21].

We triangulated data from these three sources (desk review, key informant interviews, and DHIS2 data) to conduct a quantitative-qualitative explanatory synthesis exploring if and how strategies mitigated or prevented drops in primary healthcare EBIs and whether they were adapted or newly implemented. We conducted a similar analysis of barriers and facilitators.

## Results

### Health system disruptions

Based on the DHIS2 public health facility data (Table 1), nationally, we found there was a 12% drop in pentavalent vaccine doses administered (IRR = 0.88; 95%CI: 0.87, 0.88), and an 11% drop in measles rubella (MR 1) vaccine doses administered (IRR = 0.89; 95%CI: 0.88, 0.89) between March to December 2020 of the period of COVID-19 in Bangladesh compared to the period of 2019 (Fig 2). Within that time period there was a sharp drop and then a period of recovery, accompanied by increased variability from month to month. Adjusted for seasonality the segmented regression analysis showed weak evidence of a small drop in vaccine doses administered (Table 1).

For maternal health services (Table 1; Fig 3), there was a marked, 49% drop in the number of pregnant women who attended four or more antenatal care visits (ANC 4) (IRR = 0.51; 95%

**Table 1. Interrupted time series analysis of Poisson regression model of the impact of COVID-19 on primary healthcare services in Bangladesh.**

| Primary healthcare services | Crude Incidence Rate Ratio (IRR) | 95% CI | Adjusted Incidence Rate Ratio (IRR)¥ | 95% CI | P-value |
|---|---|---|---|---|---|
| **Childhood vaccination** | | | | | |
| Pentavalent vaccine doses | 0.876 | 0.873, 0.879 | 1.021 | 0.773, 1.350 | 0.880 |
| Measles-Rubella doses (MR) | 0.890 | 0.887, 0.893 | 0.981 | 0.740, 1.300 | 0.891 |
| **Maternal health services** | | | | | |
| 4 or more antenatal care (ANC) care visits | 0.514 | 0.509, 0.520 | 0.626 | 0.516, 0.759 | p<0.001 |
| Facility-based delivery | 0.588 | 0.582, 0.594 | 0.797 | 0.673, 0.943 | 0.008 |
| Postnatal care | 0.587 | 0.582, 0.592 | 0.806 | 0.664, 0.977 | 0.028 |
| **Outpatient attendance for children** | 0.443 | 0.442, 0.444 | 0.469 | 0.312, 0.706 | p<0.001 |

Adjusted Incidence Rate Ratio¥: adjusted for seasonality

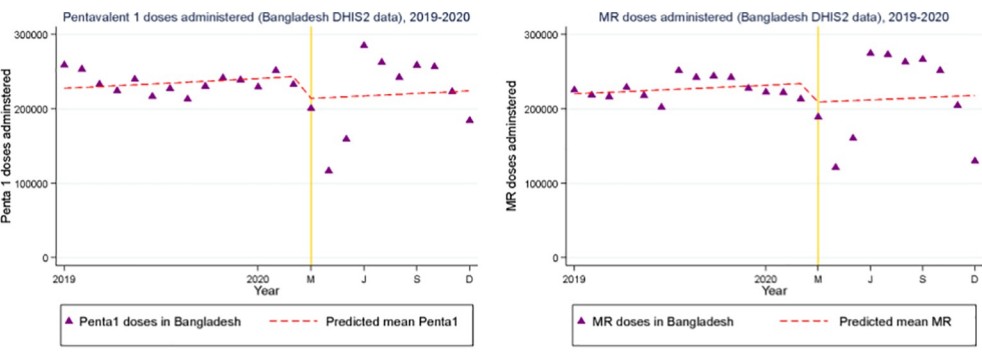

**Fig 2.** Interrupted time series with level change regression model of (a) pentavalent 1 and (b) measles rubella doses of vaccine administered in Bangladesh from 2019 to 2020. Yellow line: First case of COVID-19 reported.

CI: 0.51, 0.52), a 42% drop in facility-based delivery (IRR = 0.58; 95%CI: 0.58, 0.59) and a 41% drop in the number of women who received postnatal care (PNC 1) services at facilities (IRR = 0.59; 95%CI: 0.58, 0.59) between March to December 2020, during the period of COVID-19. Similarly, there was a marked drop (about 56%) in outpatient department attendance for children under 5 in public health facilities nationally (IRR = 0.443; 95%CI: 0.442, 0.444) between March to December 2020 as compared to the period of 2019 (Table 1). We found that these EBIs began recovering starting in June 2020.

The results from the key informant interviews were consistent with the DHIS2 interruption data and in line with the findings of the literature review (S1 Table). The KIIs additionally provided information on other services, factors, and strategies that were put into place during this phase of the COVID-19 pandemic.

## Contextual factors

We found that the COVID-19 pandemic and the response to the pandemic resulted in the emergence of a number of contextual factors, mirroring to some extent what was happening globally, which were challenging to EBI delivery. Key informants and our literature review identified factors that were new during the COVID-19 pandemic and that compromised

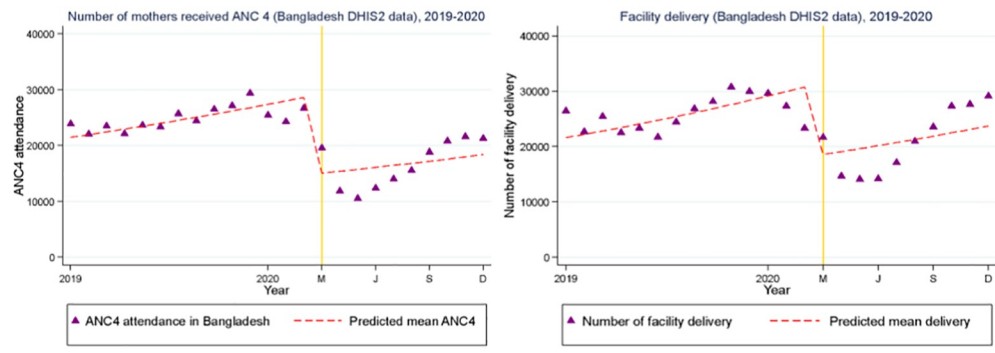

**Fig 3.** Interrupted time series with level change regression model of number of pregnant women who (a) received antenatal care (ANC 4) services and (b) had facility deliveries in Bangladesh from 2019 to 2020. Yellow line: First case of COVID-19 reported.

Bangladesh's ability to maintain the delivery of essential health services including for under-5-focused primary healthcare EBIs. On the demand side, important barriers to maintaining essential health services included fear of infection at health facilities, transportation challenges, lockdown and movement restrictions, and difficulties accessing medical care resulted in the reduction of health-seeking behavior and service utilization. Barriers from the supply side included the redirection of the health system's focus to the COVID-19 pandemic, workload and staff shortages including due to assignment to COVID-19 duty, panic among healthcare workers, and confusion on how to handle the novel virus (Table 2). Even prior to the pandemic, the country's health system suffered from staff shortage. This was exacerbated in the first months of COVID-19. A KI noted that they were working with a shortage of staff, and that specifically within the immunization program, 30% of the existing posts were still vacant. This not only affected access to services but also the quality of the services delivered. A KI explained,

> "The huge HR [human resources] shortage still kind of a valid factor you know that actually hampering the quality service delivery. Whenever we are working at the facility level, this issue is coming up as one of the critical barriers in terms of provision of quality of care, quality of MNCH services."

The facilitating and challenging contextual factors we identified very closely reflected what we identified during the previous case study in Bangladesh covering 2000–2015 [11]. Existing factors such as data availability, a strong community health system, and a culture of collaboration and coordination facilitated the reduction of U5M between 2000 and 2015, as well as the recovery of the coverage of EBIs starting in June 2020 (Table 2). However, we found from the KIIs that several contextual factors *both* facilitated and hindered the continued delivery of primary healthcare EBIs during the COVID-19 pandemic. For instance, the availability of data allowed the country to track the delivery of EBIs, but the lack of integration between public and private facility data prevented Bangladesh from having a full picture of the situation at hand including what levels of disruption versus care-seeking at private sector facilities was occurring. A KI explained,

> "We talk a lot about data; unfortunately, all are public data, not a single one is private data. And you will never get a total picture until the total data system is integrated and the private data system comes into light where private and public data is fifty-fifty, right?"

## Implementation strategies to prevent or respond to health system disruptions

Bangladesh implemented various strategies used to prevent or respond to disruptions in health system-delivered EBIs. Most of these strategies existed prior to the COVID-19 period, but the country implemented some of them with adaptations as needed. Through KIIs and the literature review, we identified key implementation strategies used to mitigate disruptions in EBI delivery. These included data use for decision-making, use of digital platforms including telemedicine, community engagement and education, leveraging community-based healthcare delivery. More detail on the broader set of strategies identified by key informants is described in Table 3.

**Data use for monitoring and decision-making.** The availability of data, particularly data from the public sector through the management information system (MIS) at the Directorate General of Health Services (DHIS2) and at the Directorate General of Family Planning,

**Table 2. Contextual factors identified from the interviews and whether or not they existed before the pandemic or emerged during this period, and key takeaways.**

| Contextual Factors | Type | | Status | Key Takeaways |
|---|---|---|---|---|
| | Facilitating | Challenging | | |
| Collaboration and coordination | X | | E | • Bangladesh leveraged existing network of collaborators to respond to COVID-19 and maintain EBIs<br>• Collaborators supported the creation of guidelines, contact tracing, research |
| Community fear of COVID-19 | | X | N | • Fear of COVID-19 was a significant barrier to health seeking behavior<br>• Patients delayed seeking care unless they were severely ill |
| Community health system and workers | X | | E | • Community health workers were critical to raising awareness<br>• Existing community health worker system allowed for service delivery at the household level in response to reduction at the facility level |
| Culture and beliefs | | X | E | • Contributing factor to the low health outcomes in two of the most underperforming districts in Bangladesh–Sylhet and Chittagong |
| Culture of accountability | X | | E | • Commitment of government to maintaining EBIs |
| Data availability, quality, and use | X | X | E | • Data availability allowed the government to track the coverage of EBIs and respond accordingly<br>• Lack of integration between private and public data presents an obstacle to understanding the full picture<br>Lack of data sharing from the MIS, DGHS presents challenges to data availability and quality |
| Financial factor (supply and demand factors) | X | X | E | • Economic strain caused by COVID-19 reduced individuals' earnings and the ability to pay for health services<br>• Individuals had no social safety net to rely on<br>• Funding was not described as a challenge on the supply side |
| Inequity | | X | E | • COVID-19 has likely disproportionately impacted hard to reach areas that were already underperforming prior to the pandemic<br>• Sylhet and Chittagong districts were provided as prime examples–districts that generally show the lowest health outcomes |
| Healthcare workers' fear of COVID-19 and infection | | X | N | • High rates of COVID-19 infection and death among healthcare workers as well as abuse and ostracization from the community<br>• Novelty of the virus and lack of training contributed to healthcare workers' fear of COVID-19 that in turn affected their work<br>• Fear and panic reduced with rowing familiarity with the virus, the availability of protocols and antigen tests up to and the enforcement of COVID-19 prevention guidelines |
| Health system structure, design, and strength | X | X | E | • Organizational structure of the health system contributed to the continuation of the delivery of services from the supply side<br>• Staff shortage noted as a significant challenge for quality healthcare delivery |
| Lockdown/ movement restrictions | | X | N | • Created an obstacle for patients that sought care at facilities and healthcare workers who provide it<br>• Such restrictions presented a challenge to in-person healthcare worker trainings, orientation, social and behavior change communication activities and delivery of health services |
| Health system ownership | X | | E | • Ownership of guidelines–WHO recommendations adapted to the local context |
| Preparedness | | X | E | • Bangladesh's health system was not prepared to respond to COVID-19 –potentially due to its clinical orientation rather than a public health one |
| National leadership and accountability | X | | E | • Leadership commitment to maintaining EBIs at all levels<br>• Collaboration between ministries facilitated response to COVID-19 |
| Supply chain system | X | X | E | • Conflicting information regarding strength of the supply chain system–some noted the strength of the supply chain system prevented stockout while others claimed that the weaknesses were exposed during the pandemic |
| System of learning and improvement | X | | E | • Government informed by real time data analysis<br>• Regular feedback and monitoring meetings to track progress and improve strategies |
| Workload and staff shortage | | X | N | • Focus of the health system and healthcare workers turned to COVID-19<br>• Increased workload shifted the attention away from ordinary EBIs and even impacted the health system's capacity to analyze data<br>• Staff shortage was not new to the COVID-19 period |

(Note: E = Existing, N = new)

DGHS: Directorate General of Health Services; EBI: evidence-based intervention; MIS: Management information system

**Table 3. Implementation strategies used to prevent and/or respond to disruptions in evidence-based interventions and key takeaways.**

| Strategies | New (N), adapted (A), or continued (C) strategy | Key takeaways |
|---|---|---|
| Bangladesh's response to COVID-19 | N | • Provision of adequate supply of personal protective equipment and infection prevention and control training<br>• Rapid formation of the National Technical Advisory Committee<br>• Availability of government fund for COVID-19 response and support to maintain EBIs<br>• Availability of separate COVID-19 and non-COVID-19 treatment centers, and allocation of healthcare workers to COVID-19 response<br>• Availability of COVID-19 rapid antigen tests at lower level including upazila health complex for triaging |
| Re-design and health system change to mitigate EBIs drop | N | • Preparation of Standard Operating Procedure (SOP) for service delivery<br>• Designated a separate queuing zone e.g., separate queues for ANC zone so that nobody could mix with each other<br>• Continued provision of facility delivery and newborn care following the COVID-19 protective measures<br>• Developed specific guideline for COVID-19 infected children and Kangaroo Mother Care Service |
| Data use for monitoring and decision-making | A, C | • Review of MIS data for service disruption, track progress, service utilization and coverage<br>• Weekly monitoring meetings organized by the DGHS including Civil Surgeons and Upazila Health and Family Planning Officers to review data and feedback to areas of low coverage<br>• Review of monthly data and provision of feedback during supervision (either through workshop or virtual meetings)<br>• Real time data analysis to inform policymakers<br>• Use of disaggregated analysis by administrative divisions |
| Use of digital platform and e-health | A, C | • Virtual meetings and trainings using Zoom, Skype, MS Teams<br>• Use of telemedicine including phone consultations<br>• Use of Viber groups for information sharing and communication<br>• Development of Facebook page, YouTube channel for sharing videos and messages |
| Community engagement and education | C | • Campaigns to raise public awareness about service provisions (e.g., with infection prevention and control), increase community awareness regarding facility delivery<br>• Initiating courtyard meetings to increase health seeking behavior so that women started taking up ANC<br>• Using media and publicity including daily briefing related to COVID-19 situation<br>• Awareness creation programs were being aired through satellite TV channels<br>• Use of COVID-19 related bulletin to educate the community<br>• Services were advertised through a Facebook page, YouTube channel and posters.<br>• Posters were put up in vaccination centers, houses, and various rural areas all over Bangladesh regarding continuity of EBIs (e.g., immunization services) |
| Guideline development | N, A | • Development of various MNCH guidelines for essential service delivery in the context of COVID-19 |
| Training provisions | A, C | • Healthcare worker training on the transmission of COVID-19<br>• Trainings on continuity of EBIs<br>• Training and orientation on the guideline conducted at all levels |
| Engagement and coordination of implementing partners and donors | C | • In collaboration with implementing partners and donors, for direct COVID-19 response such as strengthening oxygen capacity building<br>• Contributing to the National Technical Working Groups<br>• Provide technical assistance e.g., supported strengthening of MIS |
| Mentorship and supervision | A | • Online monitoring and feedback sessions<br>• Use of e-tracking system for vaccine programs |
| Leveraging community-based healthcare delivery | C, A | • Restoration of outreach sessions for immunization, and strengthening immunization services in urban slums through joining the city corporation<br>• Household visits to raise awareness about COVID-19 and EBIs<br>• Health workers provided education to pregnant mothers and children over the phone<br>• Strengthening community clinics and conducting syndromic surveillance using the community support team |
| Focus on equity | C, A | • Government of Bangladesh focused on hard-to-reach areas, and areas with predominant social taboo<br>Allocation of funds for hard-to-reach areas (e.g., for EPI program activities) |

*(Continued)*

**Table 3.** (Continued)

| Strategies | New (N), adapted (A), or continued (C) strategy | Key takeaways |
|---|---|---|
| Provision of transport | A | • Some provision of transport for healthcare workers |
| Human resources strengthening | A | • Gradual increase in human resources deployment by the government; additional recruitment of health workers<br>• Rotation of healthcare workers |
| Leveraging existing experience | C | • Utilizing previous experience and capacity (e.g., previous preparation for EBOLA, SARS) to leverage COVID-19 response |

EPI: Expanded Programme on Immunization; MIS: management information system; MNCH: Maternal, newborn, and child health

allowed the MoH&FW to track the coverage of EBIs and respond accordingly. A number of KIs emphasized the importance of Bangladesh's use of data for decision-making and system of learning. Some specific implementation strategies utilized during the period of COVID-19 to respond or prevent to EBI disruptions related to data use included weekly monitoring meetings related to Expanded Programme on Immunization organized by the DGHS, use of public MIS data to track progress at the district level or upazila (subdistrict) level, service utilization and coverage, prediction of frequency of death using Sample Vital Registration System data, review of monthly data and provision of feedback during supervision, and review of data for service disruption including children who missed vaccination. According to one KI,

> "When we saw these child health services, . . . are going down, immediately we had this mechanism to sit with the program people, here at the national level, and. . .they identified the priority actions. We supported them with data because this is the kind of capacity they require."

**Use of digital platform and e-health.** Bangladesh began implementing digital health services to promote and protect public health, including telehealth, video consultation, data collection from remote rural areas, monitoring, surveillance, and human resource development including continued professional development, well before the COVID-19 pandemic. During the pandemic, additional services were added including risk communications, contact tracing and hot spot identification, and strengthening of telemedicine and other pre-existing services. This move towards digital services worked as a bridge between providing safe healthcare consultation while keeping social distance as advised [22]. One KI said:

> "We developed a website from NNHP [National Newborn Health Program] and IMCI [Integrated Management of Childhood Illness]. We created it in a way where [users] can contact us or. . .get the name and mobile number of the head of the upazila health complex in that area. We also developed videos on how our services work in COVID-19 situation. . . The direct telemedicine service and the emergency telemedicine service was running as usual in our hospitals throughout the COVID-19 situation. Actually, we did not launch any different telemedicine services during this period."

**Community engagement and education.** Key informants described community fear of COVID-19 in Bangladesh as a significant barrier to health seeking behavior during the pandemic. Various community engagement and awareness formation strategies were used to alleviate fear and ensure continuity of EBIs. One KI explained, *"Our health workers have made household visits. We have about 50,000 health workers. They, along with our other 50,000 field*

*staff of Bangladesh Rural Advancement Committee [BRAC], conducted household visits and raised awareness about the COVID-19.”*

**Leveraging community-based healthcare delivery.** Bangladesh's community health workers comprised multiple cadres which were used to implement a number of interventions. A KI explained that *“This is a strategic approach where we have strengthened our community clinics on how to respond to such crises and we capacitate them to respond to this crisis. We did this along with the government.”* Specific strategies, used particularly by BRAC, included restoration of outreach sessions for immunization, household visits to raise awareness about COVID-19, health worker-provided education to pregnant mothers and children over the phone, strengthening immunization through joining the city corporation in urban slums, syndromic surveillance using the community support team in urban areas, and strengthening community clinics in terms of triaging, providing personal protective equipment, and provision of infection prevention and control and data management training to staff.

## Implementation outcomes

Different implementation strategies which were adapted or newly implemented during the pandemic were associated with a number of implementation outcomes including acceptability, reach or coverage, fidelity, appropriateness, feasibility, equity, and effectiveness. Key informants reported that the country had disruptions to EBI delivery in the early months of the COVID-19 pandemic. For instance, childhood vaccination coverage was around 80% before the pandemic but fell below 50% when the pandemic emerged in the country. With effective collaboration and engagement with stakeholders, this immunization coverage rose to around 90%, reflecting effectiveness of the strategy. Table 4 summarizes implementation strategies with their associated implementation outcomes.

## Resiliency findings

Bangladesh's health system demonstrated resilient characteristics due to previous work to strengthen data systems, establish collaborative networks, and adopt technologies that could be adapted as service delivery needs changed. The health system was able to adapt service delivery through the adoption of technology such as telemedicine to address the reduction in service utilization at the facility level. Moreover, it was able to leverage existing collaboration networks to respond to COVID-19 and to maintain primary healthcare EBIs. The existing data system also allowed the health system to track EBI availability and use, and respond to drops or prevent further disruption with targeted strategies.

## Key transferable lessons

A number of lessons emerged from the KIIs for recommendations for other countries or contexts stemming from their experience in the first phase of the COVID-19 pandemic (S2 Table). The largest number of respondents (six) identified “building strong collaboration and a strong system of coordination of activities between the MoH&FW, donors, and implementing partners including multi-sectoral collaboration supported the implementation of EBIs during the pandemic” as an important lesson. According to one KI, *“Proper arrangements and the continuation of good collaboration between the state apparatus or our administration or the inter-NGO sector or INGOs with our health department is the key. And in this time of crisis, everyone has to come forward.”*

“Invest in health systems, inputs, and quality” was also described by a high number of respondents (five). Most KIs reported the presence of a strong structure of the health system and investing in diverse health system inputs and quality at all levels of care (central, district,

**Table 4. Implementation strategies and outcomes.**

| Implementation outcome | Implementation strategy | Mechanism |
|---|---|---|
| Feasibility | Leveraging pre-existing strong data management system | This system allowed to detect disruptions in the delivery of EBIs:<br>"... we were doing some kind of analysis since this pandemic came in ... from the very beginning we tried to kind of look into the data from the routine HMIS ... we have a very strong data in the DHIS2 that is the official MIS, so we tried to look at all these maternal, newborn and child health particularly the child health indicators there. And we found that there has been some kind of disruption in the services. So, we tracked all along this during this COVID period ..." |
| Appropriateness | Updating clinical guidelines with COVID-19 preventive measures<br>Using telemedicine for the delivery of COVID-related services | The guidelines helped staff in healthcare facilities to keep delivering EBIs, resulting in fewer disruptions:<br>"...An interesting thing that I found while looking at the data is that the service disruption had happened everywhere, but it was relatively a little better in community clinics. It was better in case of service delivery or care seeking at the community clinic than the upazila health complex. That is the overall scenario I got..."<br>Telemedicine supported service delivery with reduced physical contact:<br>"... We have launched another telemedicine service. It is just a telemedicine service of the doctors and our health workers although it was only made for COVID cases ... We saw that the telemedicine service we introduced during the COVID period for COVID interventions helped people a lot in terms of seeking COVID related services. The reason is that many people actually call together in 333 or 16263 telemedicine services ... the response we got from the localized telemedicine service was much better ... Now there is no need for physical service for everything ..." |
| Fidelity | Training on updated clinical guidelines | After orienting health providers to the updated clinical guidelines, the delivery of EBIs gradually continued:<br>"... after developing the guideline, we had to inform the service providers about it. Next, we virtually joined our service providers at different levels ... and gave them an orientation about our guideline. Yes, after the orientation they started trying to continue the service according to our guideline and gradually started working."<br>Another KI reported that care services were delivered as usual:<br>"... In the pandemic condition we are keeping our services as it was before, like, when a child comes to IMCI corner, we will do the triaging first. We brought the service in line on how should we treat a child in IMCI Corner if he is non-COVID or COVID affected ..." |
| Acceptability | Community education about not fearing to receive EBIs from health facilities due to COVID-19 infection | As health providers were not aware of infection prevention and control (IPC) practices, IPC training was provided to ensure that the delivery of EBIs respects COVID-19 preventive measures:<br>"... We had 26,500 staff as health assistants (HA) in Bangladesh and we realized that they did not yet know how to prevent themselves from infection in healthcare settings. So, we gave the IPC training to each of them ... we also developed some posters on this, and these posters were put up in vaccination centers, houses and various areas all over in the rural areas of the country. In the posters, 'how the parents would bring the children for vaccination during this COVID-19 situation, and they must not omit vaccinating' was explained. This matter of inspiration was there ..."<br>After the community members were told not to fear COVID-19 and to seek care at health facilities, they started taking up the services:<br>"... We started telling the parents again that you should go to any upazila health complex or hospital where the services are now available. And we started providing these services, especially, nutrition, ARI, diarrhea and all other services that we give to children, and ANC, PNC. So, it was observed that when the people was benefitted, they started taking these services with great interest since they needed it very much ..." |
| Effectiveness | Collaboration and engagement with different stakeholders | When EBIs were disrupted during the months of March, April, May, and June, different stakeholders worked together to bring EBI delivery back to normal:<br>"... maternal, newborn and child health services was seriously disrupted in the month of March, April, May and June. Then what we did actually, the stakeholders responded rapidly that how we can come back on the track and here the Ministry of Health and Family Welfare took a real initiative, particularly if I say MNCAH, that the DGHS, MNH as well ... so, they came forward ... and rapidly we accumulated the evidence from WHO and others and prepared ourselves... the EPI program with which we were always happy and being appreciated around the globe, that was also disrupted. And I remember that the 80+ vaccination rate, in some vaccines 90+, came down below to 50%. So, from that higher level when it comes below 50, it's not only a fall, it's a dramatic fall. And then it has raised again to 80% in some of the vaccination centers, closer to 90% in some areas ... we have motivated our vaccinators..." |

*(Continued)*

**Table 4.** (Continued)

| Implementation outcome | Implementation strategy | Mechanism |
|---|---|---|
| Reach/ coverage | Cascaded training of health workers on COVID-19 related clinical protocols<br>Effective communication | The training was given to a considerable number of health workers:<br>"... At first we gave this infection prevention training to about 1200 Managers through Zoom. Initially we formed the Master Trainers ... we gave this training to the 6,000 new doctors who have been recruited recently. We sent them to the upazila health complexes...The Master Trainers have been trained in the first 3 batches. Then these Master Trainers trained those 6,000 doctors...and they were spread to different places at the district level and after returning back to their own districts, they trained the field staff there..."<br>During the pandemic, Bangladesh provided effective communication to community through media and reached people from different corners of the country:<br>"... all the media of publicity and all the conventional media were being used. We know that the electronic media actually reach people very quickly ... every day a briefing was given regarding the status of the country's COVID situation. That briefing was given from the DGHS ... people from different corners of the country waited ... huge people listened it every day with interest ... our main focus was to give an update on the current status but, simultaneously, the message was conveyed through this way too many people." |
| Equity | Focus on equity | The inequity in EBI coverage that had been a challenge in some divisions continued to pose a challenge during the pandemic:<br>"... Of course, we have geographical disparity or inequality ... our main area of concern is Sylhet and Chittagong Division ... In Sylhet and Chittagong, the people are quite affluent. But the social taboo and care seeking pattern are not upright there. They suffer much from social taboo especially in Sylhet; the entire Sylhet Division and the hill tract areas in Chittagong. Another thing is that there are plenty of hard-to-reach areas in one or two divisions ... it was difficult to deliver or enforce interventions in those areas over the years and getting the results was hard too...and still is ...the inequality still persists and maybe will persist too since the care seeking pattern will not change overnight in those places. Though changes are happening, but a hard-to-reach area does not become reachable overnight ... Their care seeking pattern and the religious taboo will not change suddenly those who live in a tea garden or in a remote area of Sylhet. It will take years and years to make these changes happen." |

DGHS: Directorate General of Health Services; EBI: Evidence-based intervention; EPI: Expanded Programme on Immunization; IMCI: Integrated management of childhood illness; MIS: management information system

community clinics, and outreach center) were critical to respond or mitigate disruptions of EBIs stemming from the COVID-19 pandemic. Using immunization as an example, a KI explained that,

> "Our immunization system is very much structured from the national level to a very outreach center. There are 24 centers in a ward, and it is operated there every month. There has never been a shortage of vaccine supply chains in these centers during the pandemic period even during more crisis periods. No problem has been faced in the vaccine supply chain at a distribution center, i.e., supply from the center to that very community level."

## Discussion

We found that Bangladesh had an initial drop in U5M-oriented primary healthcare EBIs during the early phase of the pandemic, and began recovering in June 2020. Demand-side barriers including fear of infection at health facilities, lockdown and movement restrictions, and difficulties accessing medical care, combined with supply-side barriers including redirection of the

health system's focus to the COVID-19 pandemic, assignment to COVID-19 duty, and confusion on how to handle the novel virus, resulted in the reduction of health-seeking behavior and service utilization. Key strategies used to prevent and respond to disruptions included data use for decision-making, use of digital platforms, community engagement, and leveraging community-based healthcare delivery. Key transferable lessons included building in collaboration and strong coordination of activities, community, and civil society engagement, and investing in health systems, inputs, and quality.

Our findings that care-seeking and access declined and then began to rebound after a period of several months were consistent with broader trends identified in the first phase of the pandemic [23, 24]. The biggest decline in care-seeking for sick children under 5 was observed between April to July 2020 –during and after the country's first national lockdown–a 70% decrease compared to the same period in 2019. However, the second half of 2020 (August to December) saw a 23% increase in care-seeking compared to the same time in 2019 [5]. Similarly, researchers in India identified substantial decreases in facility-based delivery (45%) in a tertiary care center, and described lockdown and movement restrictions, as well as fear of COVID infection, to have been important barriers in care-seeking [25]. Figs 2 and 3 showed fluctuating trends immediately before the COVID-19 period, with lower vaccines doses administered and lower facility deliveries in the month or months before the lockdown began. These slight declines just before the official report of COVID-19 in the country could be due to the initial disinformation and media campaign related to COVID-19 globally even prior to the first cases of COVID-19 confirmed on March 8, 2020; however, issues of data quality including missing data and reporting errors in the DHIS2 dataset are difficult to exclude. Some amount of seasonal variability likely also explains some of the trend.

The barriers described by the key informants largely concurred with those reported globally [26, 27]. For instance, in both WHO Pulse Survey reports, countries noted challenges including reduced outpatient care attendance, reduced access to healthcare services due lack of transport during lockdowns, financial difficulties, and fear [26, 27]. On the supply side, some countries reported cancellation and unavailability of services, overburdening of healthcare staff capacity due to re-deployment to COVID-19 relief, insufficient personal protective equipment for healthcare workers, and supply chain difficulties. Bangladesh also faced these challenges with variable magnitude and response; service utilization decreased overall but gradually recovered. This recovery was due to improvement on both the demand and supply side.

Bangladesh's ability to use and adapt existing implementation strategies to face new barriers specific to the COVID-19 pandemic indicates health system resilience [20]. The country adapted service delivery through adopting technology such as telemedicine to address reduced service utilization at the facility level. It leveraged existing collaboration networks to respond to COVID-19 and to maintain EBIs; the existing data system allowed the health system to track EBI availability and use, and respond to drops or prevent further disruption with targeted strategies. Examples like these suggest that existing facilitating contextual factors such as collaboration and coordination, built before the pandemic, were critical to societal resilience in maintaining delivery of services during the pandemic [20, 28]. However, challenging contextual factors such as the pre-existing staff shortage undermined the resiliency of the health system. As staff were redirected to the COVID-19 response, this worsened staff shortages, further affecting service delivery and quality. As such, the health system was not able to fully withstand the shock of the pandemic on the delivery of EBIs. The recovery of services after their initial drop during the lockdown in late March took a period of several months.

The findings of this study help to underline the importance of adaptation of existing approaches to unique but increasingly common health system shocks such as the COVID-19

pandemic or future health threats. The use of the already existing community health system and structure to adopt new implementation strategies like digital technologies to improve service availability, and to adapt existing implementation strategies such as community engagement and education to reduce fear and increase care-seeking, argue for the importance of strengthening healthcare systems in periods between crises in order to increase resilience during shocks [29]. Future research in this subject would benefit from the inclusion of community representation in key informant interviews, which was not possible for this work due to time and resource constraints, and which would have provided valuable insight from the demand side in building knowledge to reduce resilience research gaps [30]. We conducted the key informant interviews during a period of surging COVID-19 cases in Bangladesh, which may have shortened the themes of what could be ideally covered in future work. In addition, the ability to detect transient changes in EBI service delivery and coverage was limited by the shorter duration of the period of COVID-19 under investigation and would be a valuable direction to pursue further.

Limitations of this study include our use of the DHIS2 dataset which is only from public facilities; community level data and data from private facilities are not available in the DHIS2 dataset. The absence of private facility data in our analysis, to some extent, prevented us from drawing a full picture of maternal, neonatal, and child health service disruptions in Bangladesh. Moreover, issues of data quality including missing data and reporting errors in the DHIS2 dataset are difficult to exclude. Our study was further limited in that community representation in our key informant interviews was not possible due to time and resources constraints, and would have provided valuable insight from the demand side. The ability to detect transient changes in EBI service delivery and coverage is limited by the shorter duration of the period of COVID-19 under investigation, as well as the 14-month duration of time prior to COVID-19 used to establish the secular and seasonal trends. We did not include analysis of subnational variability of EBI delivery.

## Conclusion

This study explored whether and how health system-delivered EBIs targeting amenable U5M were maintained during the first phase of COVID-19 in Bangladesh. We identified strategies used to prevent or respond to EBI drops, the factors that helped or hindered the response to COVID-19-related threats to EBI uptake and delivery, and how the previous efforts to implement these EBIs between 2000–2015 supported the work to maintain EBIs during the pandemic in Bangladesh and contributed to resiliency. Countries working to increase EBI implementation can learn from the barriers, strategies, and transferable lessons identified in this work in an effort to reduce and respond to health system disruptions in anticipation of future health system shocks.

## Supporting information

**S1 Checklist. Standards for Reporting Implementation Studies (StaRI) Checklist.**
(PDF)

**S2 Checklist. COnsolidated criteria for REporting Qualitative research (COREQ) Checklist.**
(PDF)

**S1 Fig. Implementation research framework for understanding evidence-based interventions to reduce under-5 mortality.**
(TIF)

**S1 Table. Composition of key informants interviewed.**
(DOCX)

**S2 Table. Transferable lessons.**
(DOCX)

## Acknowledgments

The authors would like to thank the key informants and other stakeholders in this work who provided information, perspective, and feedback on our findings.

## Author Contributions

**Conceptualization:** Alemayehu Amberbir, Agnes Binagwaho, Lisa R. Hirschhorn.

**Data curation:** Alemayehu Amberbir, Kedest Mathewos, Jovial Thomas Ntawukuriryayo.

**Formal analysis:** Alemayehu Amberbir, Amelia VanderZanden, Kedest Mathewos.

**Funding acquisition:** Agnes Binagwaho, Lisa R. Hirschhorn.

**Investigation:** Alemayehu Amberbir, Fauzia A. Huda, Amelia VanderZanden, Kedest Mathewos, Jovial Thomas Ntawukuriryayo, Agnes Binagwaho, Lisa R. Hirschhorn.

**Methodology:** Alemayehu Amberbir.

**Project administration:** Kedest Mathewos, Agnes Binagwaho, Lisa R. Hirschhorn.

**Supervision:** Alemayehu Amberbir, Fauzia A. Huda, Agnes Binagwaho, Lisa R. Hirschhorn.

**Writing – original draft:** Alemayehu Amberbir, Amelia VanderZanden.

**Writing – review & editing:** Alemayehu Amberbir, Fauzia A. Huda, Amelia VanderZanden, Kedest Mathewos, Jovial Thomas Ntawukuriryayo, Agnes Binagwaho, Lisa R. Hirschhorn.

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
