## [Decision Letter · Decision Letter 0]

10 Nov 2023

PGPH-D-23-01582

Mitigating the impact of COVID-19 on primary healthcare interventions for the reduction of under-5 mortality in Bangladesh: lessons learned through implementation research

Dear Dr. Amberbir,

Thank you for submitting your manuscript to PLOS Global Public Health. After careful consideration, we feel that it has merit but does not fully meet PLOS Global Public Health’s publication criteria as it currently stands. Therefore, we invite you to submit a revised version of the manuscript that addresses the points raised during the review process.

We look forward to receiving your revised manuscript.

Kind regards,

Orvalho Augusto, MD, MPH

Academic Editor

Journal Requirements:

Additional Editor Comments (if provided):

This a very well written manuscript and leveraging available data from the routine information system and mixed methods to enrich the understanding of COVID-19 impact in Primary Health.

There are, however, few issues as pointed below by the 2 reviewers.

Reviewers' comments:

Reviewer's Responses to Questions

**Comments to the Author**

1. Does this manuscript meet PLOS Global Public Health’s publication criteria? Is the manuscript technically sound, and do the data support the conclusions? The manuscript must describe methodologically and ethically rigorous research with conclusions that are appropriately drawn based on the data presented.

Reviewer #1: Partly

Reviewer #2: Yes

2. Has the statistical analysis been performed appropriately and rigorously?

Reviewer #1: I don't know

Reviewer #2: No

3. Have the authors made all data underlying the findings in their manuscript fully available (please refer to the Data Availability Statement at the start of the manuscript PDF file)?

Reviewer #1: No

Reviewer #2: No

4. Is the manuscript presented in an intelligible fashion and written in standard English?

Reviewer #1: Yes

Reviewer #2: Yes

5. Review Comments to the Author

Reviewer #1: - The manuscript is difficult to read and would be best to re-write and make it concise

- The methodology of quantitative need to be explained more how to get the data of EBI? is it simply collected by the MOH data? Why is it need to combine with the "key informants" data

- IN Figure 1, why the Penta and MR dose higher in 2020 compare to 2019

- This analysis need to be linked to the time line data of COVID-19 lockdown or community restriction

Reviewer #2: This outstanding study documented the disruptions inflicted by COVID-19 on primary healthcare services, while also shedding light on the mitigating measures employed. Utilizing a mixed methods approach, the authors thoroughly investigated the issue from multiple perspectives, delivering a comprehensive and nuanced comprehension of its impact.

I have a few suggestions:

1. On page 8, the impact of COVID-19 was illustrated by comparing the period from March to December 2020 to the corresponding period in 2019. However, considering the rapidly evolving situation of COVID-19, its impacts on primary card, and the dynamic nature of mitigating measures, it would be more engaging to conduct a month-to-month comparison. While some monthly data is available in later sections, such as page 17, a systematic presentation of these monthly comparisons would enhance the study's main points.

2. I could not find the “Poisson regression model to estimate the impact of COVID-19 on DHIS2 indicators” mentioned the study design. I recommend that the authors provide a clear and explicit explanation of the model's specifications and present the regression results for better clarity and understanding.

3. Figures 1 and 2 display intriguing temporal trends. Figure 2 effectively illustrates the recovery of antenatal and delivery services following the initial significant disruption. In the case of Figure 1, it underscores the initial dramatic disruption, but the subsequent trajectory requires more in-depth exploration and explanation. Understanding why there was a sudden recovery, followed by a deterioration in the last few months, is crucial. To provide further clarity, it would be beneficial to incorporate a timeline of major mitigating measures, as this may shed light on the fluctuations in these trends.

6. PLOS authors have the option to publish the peer review history of their article (what does this mean?). If published, this will include your full peer review and any attached files.

**Do you want your identity to be public for this peer review?** For information about this choice, including consent withdrawal, please see our Privacy Policy.

Reviewer #1: **Yes: **NINA DWI PUTRI

Reviewer #2: No

---

## [Decision Letter · Decision Letter 1]

23 Jan 2024

PGPH-D-23-01582R1

Mitigating the impact of COVID-19 on primary healthcare interventions for the reduction of under-5 mortality in Bangladesh: lessons learned through implementation research

Dear Dr. Amberbir,

Thank you for submitting your manuscript to PLOS Global Public Health. After careful consideration, we feel that it has merit but does not fully meet PLOS Global Public Health’s publication criteria as it currently stands. Therefore, we invite you to submit a revised version of the manuscript that addresses the points raised during the review process.

We look forward to receiving your revised manuscript.

Kind regards,

Orvalho Augusto, MD, MPH

Academic Editor

Journal Requirements:

Additional Editor Comments (if provided):

This is a well-written implementation research report to document the challenges and accomplishments in mitigating the COVID-19 impacts in the provision of a few essential child healthcare services in Bangladesh. It is one of few examples combining both qualitative and quantitative methods.

One issue: There are only 14 months prior to COVID-19 to establish the secular and seasonal trends. This deserves a comment on the limitations.

Reviewers' comments:

Reviewer's Responses to Questions

**Comments to the Author**

1. If the authors have adequately addressed your comments raised in a previous round of review and you feel that this manuscript is now acceptable for publication, you may indicate that here to bypass the “Comments to the Author” section, enter your conflict of interest statement in the “Confidential to Editor” section, and submit your "Accept" recommendation.

Reviewer #3: (No Response)

2. Does this manuscript meet PLOS Global Public Health’s publication criteria? Is the manuscript technically sound, and do the data support the conclusions? The manuscript must describe methodologically and ethically rigorous research with conclusions that are appropriately drawn based on the data presented.

Reviewer #3: Yes

3. Has the statistical analysis been performed appropriately and rigorously?

Reviewer #3: No

4. Have the authors made all data underlying the findings in their manuscript fully available (please refer to the Data Availability Statement at the start of the manuscript PDF file)?

Reviewer #3: Yes

5. Is the manuscript presented in an intelligible fashion and written in standard English?

Reviewer #3: Yes

6. Review Comments to the Author

Reviewer #3: Researchers employed one of the important aspects of implementation science research to document experiences and lessons in mitigating COVID-19's impact on the delivery of evidence-based child health interventions in Bangladesh. They identified the contextual challenges and strategies the country employed and highlighted key lessons for future programming to improve the implementation of maternal and child evidence-based practices.

Comments

1. The title of the manuscript talks about the impact of COVID-19 on primary healthcare interventions. But it is not discussed in the text. It would also be good to describe Bangladesh’s primary healthcare delivery system

2. Background: I would combine the first and fourth paragraphs (lines 101-105) and present them as a problem statement. I would also move the last sentence of the third paragraph (lines 99-100) to the first paragraph.

3. Evidence showed that care-seeking for sick children rebounds later than other maternal and child health services. I was expecting care-seeking for sick children (such as pneumonia and diarrhea) as one of the key EBI for under-5 children to be explored in this study.

4. I am not sure the Poisson regression model researchers did is a true time series analysis method that would show pre-COVID, immediate, and post-COVID trends. I would report the monthly rate of decline in EBIs from the interrupted time series analysis (ITSA) output: 1) the pre-COVID-19 trend, 2) the immediate effect of COVID-19, and 3) the post-COVID-19 change (that is the differences between pre-COVID-19 and post-COVID-19 periods) on the EBI child health intervention coverage attributable to COVID-19 with counterfactual estimates of what would have happened without COVID-19, given the pre-existing trend of EBIs. For this purpose, researchers can use the “itsa” Stata command. Also, present model assumptions.

5. Figures 1 and 2 showed a decline trend before COVID-19 was first confirmed in the country. Can you comment what does it mean?

Minor comments;

Line 78, add “,” after the word pandemic; line 239, spelt out HR;

7. PLOS authors have the option to publish the peer review history of their article (what does this mean?). If published, this will include your full peer review and any attached files.

**Do you want your identity to be public for this peer review?** For information about this choice, including consent withdrawal, please see our Privacy Policy.

Reviewer #3: **Yes: **Gizachew Tiruneh

---

## [Decision Letter · Decision Letter 2]

16 Feb 2024

Mitigating the impact of COVID-19 on primary healthcare interventions for the reduction of under-5 mortality in Bangladesh: lessons learned through implementation research

PGPH-D-23-01582R2

Dear Dr Amberbir,

We are pleased to inform you that your manuscript 'Mitigating the impact of COVID-19 on primary healthcare interventions for the reduction of under-5 mortality in Bangladesh: lessons learned through implementation research' has been provisionally accepted for publication in PLOS Global Public Health.

Best regards,

Orvalho Augusto, MD, MPH

Academic Editor

Reviewer Comments (if any, and for reference):

Reviewer's Responses to Questions

**Comments to the Author**

1. If the authors have adequately addressed your comments raised in a previous round of review and you feel that this manuscript is now acceptable for publication, you may indicate that here to bypass the “Comments to the Author” section, enter your conflict of interest statement in the “Confidential to Editor” section, and submit your "Accept" recommendation.

Reviewer #3: (No Response)

2. Does this manuscript meet PLOS Global Public Health’s publication criteria? Is the manuscript technically sound, and do the data support the conclusions? The manuscript must describe methodologically and ethically rigorous research with conclusions that are appropriately drawn based on the data presented.

Reviewer #3: Yes

3. Has the statistical analysis been performed appropriately and rigorously?

Reviewer #3: Yes

4. Have the authors made all data underlying the findings in their manuscript fully available (please refer to the Data Availability Statement at the start of the manuscript PDF file)?

Reviewer #3: Yes

5. Is the manuscript presented in an intelligible fashion and written in standard English?

Reviewer #3: Yes

6. Review Comments to the Author

Reviewer #3: Thank you so much for addressing our comments and produce such a great manuscript that document Bangladesh's experiences and lessons of mitigating COVID-19 impact on maternal and child health service delivery.

7. PLOS authors have the option to publish the peer review history of their article (what does this mean?). If published, this will include your full peer review and any attached files.

**Do you want your identity to be public for this peer review?** For information about this choice, including consent withdrawal, please see our Privacy Policy.

Reviewer #3: **Yes: **Gizachew Tiruneh
